# SMixer: Rethinking Efficient-Training and Event-Driven SNNs

[1]**Yijie Lu**  [2]**Xinhao Luo**  [1]**Yixing Zhang**  [1]**Zhiyan Wang**  [3]**Wentao Li**  [3]**Yanhan Wang**
[3]**Zhi Liu**  [3,4]**Zhaokun Zhou**[*]  [2]**Guoqi Li**[*]
[1]Peking University       [2]Institute of Automation, Chinese Academy of Sciences
[3] Shandong University       [4] Tencent
zhaokunzhou@tencent.com,  guoqi.li@ia.ac.cn

## Abstract

Spiking Neural Networks (SNNs) offer a promising, energy-efficient paradigm for computation, but face challenges in performance and training costs. For example, Spiking ResNet exhibits relatively low performance, whereas high-performance Spiking Transformers are not truly event driven and cannot be implemented on asynchronous chips. Moreover, the intrinsic time steps and neuron state dynamics result in a substantial computational overhead for training SNNs. In response to these problems, we discuss rational architectural design for SNNs and argue that such designs should exhibit three key characteristics: fully event-driven operations, low training overhead and competitive performance. In light of this, we adopt the event-driven friendly Spiking-token Mixer (SMixer) as the foundational architecture and develop a spiking-feature Spatial-Temporal Pruning (STP) framework with a high pruning ratio and no trainable parameters to reduce the training overhead. Based on a statistical analysis of sparse spiking features, STP eliminates redundant spiking features across both spatial and temporal dimensions, thereby reducing the input features and computational overhead during training. Specifically, it adaptively selects the most salient spike events in the spatial domain and dynamically constrains the neurons' simulation time steps and firing thresholds in the temporal domain. By leveraging architectural design and STP, SMixer accelerates training while ensuring a fully event-driven characteristics and maintaining competitive performance, offering valuable insights for SNNs' design.

## 1 Introduction

Spiking Neural Networks (SNNs) Maass (1997), the third generation of neural networks, are distinguished by their biological plausibility Roy et al. (2019), event-driven nature, and energy efficiency. By emulating the dynamics of biological neurons, SNNs employ asynchronous binary spikes to transmit information. Consequently, a neuron's membrane potential is updated only upon spike arrival. This event-driven property allows SNNs to inherently avoid computations involving zero-valued activations, making them highly suitable for implementation on specialized neuromorphic hardware like TrueNorth Merolla et al. (2014) and Loihi Davies et al. (2018).

Despite the encouraging progress of SNNs, several challenges hinder their application in real-world pattern recognition. A fundamental difficulty lies in developing potent architectures for event-driven SNNs that not only achieve competitive performance but also adhere to the principles of purely event-driven operations for practical deployment. While Spiking Convolutional Neural Networks Lv et al. (2023) often lack sufficient performance, Spiking Transformers Zhou et al. (2023c; 2024a); Yao et al. (2025) have achieved the state-of-the-art (SOTA) results across various domains. However, a critical issue arises on asynchronous hardware where spike arrival times are less precise than in synchronous, clock-driven scenarios. The core Spiking Self-Attention mechanism, which relies on the multiplication of two spike matrices, is susceptible to significant computational deviations under such conditions Deng et al. (2024). SNNs' high training cost is also a critical issue that cannot be overlooked, primarily due to two reasons: i) Before deployment on neuromorphic chips, spiking

---

[*]Corresponding authors.

neural networks typically require training on Graphics Processing Units (GPUs) Fang et al. (2023). SNNs cannot be trained on GPUs in a truly event-driven manner, which means that even zero-valued features consume computational resources. ii) The inherent time steps and hidden states of spiking neurons further occupy computational resources. To address these challenges, we propose that rational architectural design for SNNs should embody three fundamental characteristics: **fully event-driven operations, low training overhead and competitive performance.**

Our purpose is to explore potential SNNs that simultaneously satisfy the above characteristics. Compared to Spiking Transformers, Spiking Token Mixer Deng et al. (2024) emerges as a more promising and feasible event-driven architecture that the spike-based matrix multiplication used to obtain attention weights is replaced with learnable parameters. Subsequently, we need to investigate approaches to reduce the training overhead of SMixer. The intrinsically low firing rates of SNNs result in significant network redundancy, making pruning a natural and promising approach for network lightweighting and acceleration. However, pruning SNNs presents several key challenges compared to ANNs: First, common unstructured weight pruning in SNNs usually offers limited practical acceleration and incurs additional training overhead, rendering its extension to SMixer impractical. This motivates the structured feature pruning for acceleration. Second, achieving significant acceleration necessitates a much higher pruning ratio ($\geq 0.3$) in SNNs than that in ANNs. Addressing these challenges, we begin with an analysis of the SMixer's spiking feature, Spatial-Temporal Spiking Feature Redundancy. Spiking representations are concentrated in specific spatial-temporal regions, implying that a large number of spike tokens containing low-information can be pruned, which lays the foundation for high-ratio pruning. Furthermore, a lightweight pruning strategy is needed to avoid introducing excessive pruning-related computational overhead.

We explore the potential of employing Spiking-token Mixer as a prototype within mainstream architectures and find that SMixer achieves competitive performance compared to spiking self-attention mechanisms across various architectures. To further reduce the training overhead, we propose a Dynamic Spatial-Temporal Spiking Pruning (DSTSP) framework tailored for the SMixer, which directly prunes redundant spiking feature to accelerate training while maintaining performance at a high pruning ratio. Inspired by previous studies measuring semantic information through activation values in feature maps Zagoruyko & Komodakis (2016); Ding et al. (2023), spiking feature redundancy is determined by accumulating spike event counts within specific regions, such as within a single token or a particular time step. Features with high accumulated values are considered important, whereas those with low values are regarded as redundant. The Spiking Spatial Token Pruning scores tokens by summing their spike values and ranking them adaptively based on network output. High-scoring tokens are passed to the Spiking-token Mixer encoder, while low-scoring 1ones are discarded or merged. For the temporal dimension, Dynamic Spiking Temporal Pruning dynamically reduces both the upper bound on the total number of spikes that neurons can emit and the number of time steps, thereby decreasing the latency of SMixer. By integrating the Mixer into various architectures, SMixer achieves superior performance compared to the original frameworks, demonstrating the potential as an mainstream SNN structure. We summarize the contributions as follows:

- We analyze the inherent architectural requirements for event-driven SNNs and the need for efficient training methodologies. Based on this, we propose a blueprint for a high-ratio spike feature structured pruning framework built upon the Spiking-token Mixer.
- Based on Spatial-Temporal Redundancy in Spiking-token Mixer, we develop the Dynamic Spatial-Temporal Spike Pruning framework, which integrates dynamic spatial and temporal spike feature pruning methods.
- We demonstrate that the SMixer architecture can achieve performance comparable to that of the Spikformers. Furthermore, we show that our efficient pruning framework built upon SMixer, accelerates training while maintaining performance close to the original model across various neuromorphic and static datasets.

## 2 RELATED WORK

### 2.1 DEEP SPIKING NEURAL NETWORKS

Recent advancements in Artificial Neural Networks (ANNs) have enabled significant performance improvements in SNNs, primarily through the adaptation of cutting-edge architectures from the

ANN domain. SpikingCNNs Lv et al. (2023) mix spike-form tokens with the learned weights of convolution kernels. However, its performance remains suboptimal. Spiking Transformers Zhou et al. (2023c;a); Zhang et al. (2024); Shi et al. (2024); Yao et al. (2023a); Qiu et al. (2024); Wang et al. (2025); Qiu et al. (2025); Zhou et al. (2024a); Yao et al. (2025) mix tokens with pairwise correlations weight matrix between spike-form query and key tokens and re-weights the spike-form value vectors to synthesize new token representations, which arrives the SOTA in various tasks. However, performing matrix multiplication between two spike matrices during the forward pass is incompatible with asynchronous neuromorphic hardware. In event-driven regimes, the spike arrival times lack the temporal precision, which results in significant differences in the output, ultimately precipitating a marked degradation in performance Deng et al. (2024). By jointly optimizing hardware feasibility and accuracy, STMixer Deng et al. (2024) fuses the spiking token feature with a trainable attention weight map, thereby eliminating spike-matrix multiplication. This design choice renders STMixer a significantly more practical and scalable architecture for real-world neuromorphic deployment.

## 2.2 SNN PRUNING METHODS.

The pruning methods aim to further enhance the efficiency of SNNs. Neuro-inspired strategies focus on synaptic regeneration Kundu et al. (2021) and dendritic motion Kappel et al. (2015). Qi et al. Qi et al. (2018) designed connection gates for synaptic pruning during the training process. Kim et al. Kim et al. (2022) focus on lottery ticket-based methods, integrating Iterative Magnitude Pruning Frankle & Carbin (2018) with Early-Bird tickets You et al. (2020) to identify more compact SNNs while Deng et al. Deng et al. (2023) formulate connection pruning and weight quantization as a unified constrained optimization problem, which they solve by using Spatio-temporal backpropagation and the alternating direction method of multipliers. Grad R Chen et al. (2021b) makes improvements to Deep R method by introducing a weight regeneration mechanism. RCMO-SNN Chen et al. (2023) presents an end-to-end minimax optimization approach for sparse learning. For Spikformer Zhou et al. (2023c), Zhou et al. Zhou et al. (2025) propose Spatial-Temporal Spiking Feature Pruning, and Liu et al. Liu et al. (2024) develop SparseSpikformer. Different from the above, we seek to devise pruning methods tailored to superior Spiking-token Mixer architectures, which is a direction that, to the best of our knowledge, remains entirely unexplored.

## 3 METHOD

### 3.1 SPIKING-TOKEN MIXER

The Spiking Self-Attention (SSA) involves synchronized matrix operations between spike-based queries $\mathbf{Q}$, keys $\mathbf{K}$, and values $\mathbf{V}$, whereas the Spiking-token Mixer involves asynchronous computations between weight matrices and inputs, as detailed below,

$$\mathbf{Q} = \mathcal{SN}(\text{L-BN}_{\mathbf{Q}}(\mathbf{X})), \mathbf{K} = \mathcal{SN}(\text{L-BN}_{\mathbf{K}}(\mathbf{X})), \mathbf{V} = \mathcal{SN}(\text{L-BN}_{\mathbf{V}}(\mathbf{X})), \tag{1}$$

$$\text{SSA}(\mathbf{Q}, \mathbf{K}, \mathbf{V}) = \mathcal{SN}(\mathbf{Q}\mathbf{K}^{\text{T}}\mathbf{V} * s), \tag{2}$$

$$\text{SMxier}(\mathbf{X}) = \mathcal{SN}(\mathbf{W_M}\mathbf{X}), \tag{3}$$

where $\mathcal{SN}$ denotes the spiking neuron and L-BN represents that the features pass through Linear and Batch Normalization sequentially. $s$ is the scaling factor. $\mathbf{X}$ is the spike-form input. However, calculating multiplications between spike-form $\mathbf{Q}$ and $\mathbf{K}$ is unsuitable for asynchronous hardware. In comparison, Spiking-token Mixing can serve as a substitute for the SSA. $\mathbf{Q}$ and $\mathbf{K}$ converge into a single learnable matrix $\mathbf{W_M}$ for attention weight, which is to gradually fit the attention map during the training process. In Figure 1, the overall architecture includes Spiking Patch Splitting (SPS), several encoder blocks composed of a Spiking-token Mixer module and Spiking Multilayer Perceptron (MLP), and classification head (CH).

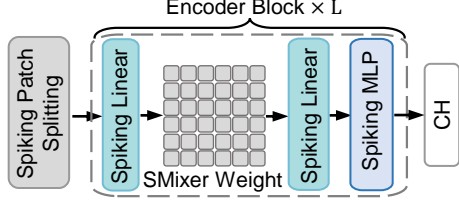

Figure 1: The overview of Spiking-token Mixer.

## 3.2 SPATIAL-TEMPORAL SPIKING FEATURE REDUNDANCY IN SMIXER

To enable efficient feature pruning in SMixer, we begin by evaluating the significance of spiking features using the Spike Intensity Value (SIV), which is defined as the sum of spike events within a designated feature region.

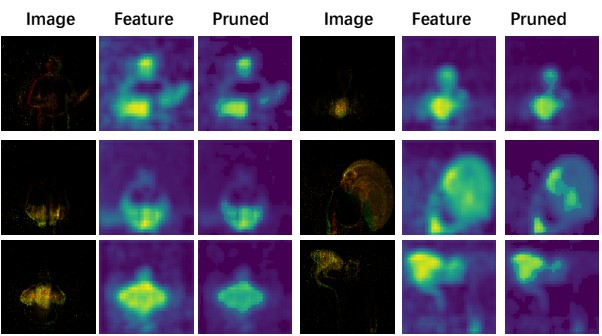

Similar to the concept of activation magnitudes in ANNs Zagoruyko & Komodakis (2016); Ding et al. (2023), a higher SIV indicates a greater concentration of semantic information. Our analysis of the DVS-Gesture reveals that the SIV distribution is highly imbalanced, with the majority of tokens exhibiting low values, as shown in Figure 3. This imbalance is closely linked to model performance: a model that uses only high-SIV tokens demonstrates considerably better accuracy than one relying on low-SIV tokens (97.9% vs 79.9%). Visual evidence further sup-

Figure 2: Visualization of Spatial Temporal Spiking Feature Redundancy.

ports this, showing that regions with high SIV align with foreground objects, while low-SIV regions tend to correspond to the background, as shown in Figure 2. It shows samples from DVS-Gesture with a spatial pruning ratio of 0.3 and a temporal pruning ratio of 0.5 The left and middle columns of each sub-figure display the original image and its corresponding original features. The right column demonstrates the result obtained by retaining feature tokens with high SIV. These findings suggest that critical semantic content is predominantly contained within a small subset of high-SIV tokens. Importantly, the computation of SIV is highly efficient, involving only a simple summation that is well-suited to the operational characteristics of SNNs.

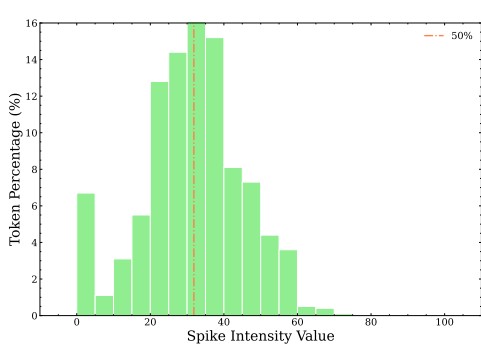

Figure 3: Distribution of Spike Intensity Value in DVS-Gesture. The maximum and minimum values are 125 and 0. The dashed line is the boundary for 50% of Spiking-tokens.

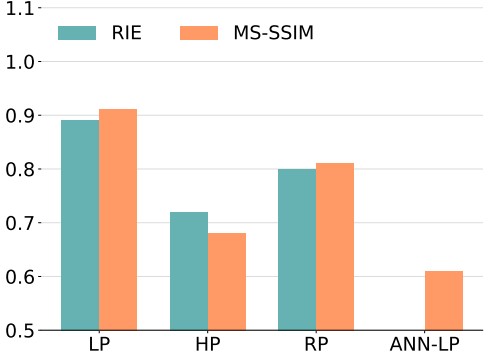

Figure 4: The strategies investigated include pruning tokens with low-SIV (LP), high-SIV (HP), and random selection (RP). ANN-LP denotes the performance of the LP strategy on the ANN-Transformer. The Y-axis denotes the metric values.

To validate the efficacy of our Spiking Information Value (SIV) metric in quantifying feature redundancy, we conducted an analysis on CIFAR-100 using RIE and MS-SSIM scores (details in Appendix C). As shown in Figure 4, we evaluated several token pruning strategies: low-SIV (LP), high-SIV (HP), and random pruning (RP), comparing pruned feature outputs against the unpruned baseline. Our analysis, which also included the LP strategy on a standard ANN-Transformer (ANN-LP), revealed two key findings. First, SIV is substantially more effective at exposing redundancy in SNNs than in ANNs. Second, the SIV-informed LP strategy is the most effective approach for SNNs. This core insight serves as the foundational principle for our DSTSP framework.

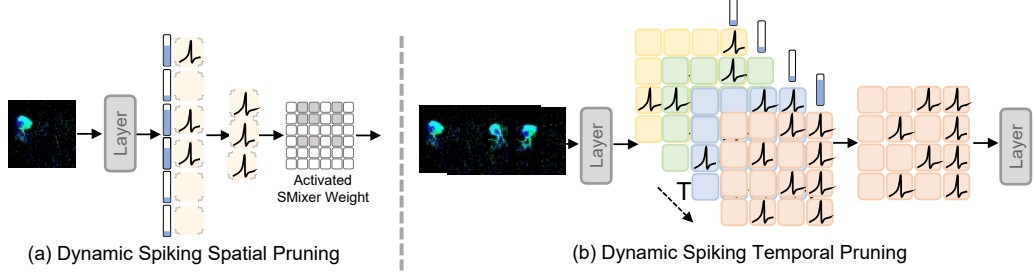

Figure 5: The illustration of Dynamic Spatial-Temporal Spiking Pruning.

## 3.3 DYNAMIC SPATIAL-TEMPORAL SPIKING PRUNING

Inspired by the high sparsity and redundancy of spiking features, we propose a lightweight method, the Dynamic Spatial-Temporal Spiking Pruning (DSTSP), to prune a significant fraction of features with low SIV, as illustrated in Figure 5. This approach aims to substantially reduce training costs while preserving inference performance. Unlike conventional pruning in ANNs, which often relies on complex, trainable modules, our method is specifically designed for SNNs. It avoids introducing any learnable parameters by leveraging simple, addition-based operations that align with the spike-driven characteristic of SNNs. DSTSP operates sequentially across two dimensions. First, in the temporal domain, it computes the SIV for each spiking feature map per time step, retaining only a specified percentage of the highest-scoring maps. Subsequently, in the spatial domain, it evaluates the SIV of individual spiking tokens, preserving only those with high activation values. The specifics of DSTSP are detailed in the following sections.

**Dynamic Spiking Spatial Pruning (DSSP)** analyzes the SIV of each token to determine its semantic information and prunes tokens with low SIV. For a spike-form feature $\mathbf{X} \in \mathbb{S}^{T \times N \times C}$, where $T$ represents the time steps, $N$ and $C$ represent the number of spiking tokens and channel dimensions, respectively. The DSSP is conducted as follows,

$$\mathbf{I_S} = \sum_{i=1}^{C} \mathbf{X}[:,:,i], \mathbf{X_i} = \text{SORT}(\mathbf{I_S}, \mathbf{X}), N' = N \cdot (1 - P_S), \mathbf{X}' = \mathbf{X_i}\left[: N'\right], \quad (4)$$

$$\mathbf{W}'_M = \mathbf{W}_M[N', N'], \text{SMixer}(\mathbf{X}) = \mathcal{SN}\left(\mathbf{W}'_M \mathbf{X}'\right) \quad (5)$$

where $\mathbf{X}$ is sorted in descending order to obtain $\mathbf{X_i}$ according to the spatial SIV $\mathbf{I_S}$. $N'$ is the number of spiking tokens after pruning and $P_S$ denotes the spatial pruning ratio. $\mathbf{X}'$ is the remained feature after DSSP. $\mathbf{W}'_M$ denotes the activated attention weight, for which we explore two approaches. The first approach treats it as a static $N' \times N'$ matrix. The second approach involves randomly selecting $N'$ shared row and column indices during training and reverting to a full-sized matrix for inference. Both strategies yield nearly identical performance at inference time.

**Dynamic Spiking Temporal Pruning (DSTP)** prunes temporal information from two aspects. On one hand, it assesses the importance of each time step to remove those that are less significant. Given an input spiking feature $\mathbf{X} \in \mathbb{S}^{T \times N \times C}$, the formulation is as follows:

$$\mathbf{I_T} = \sum_{n=1}^{N} \sum_{i=1}^{C} \mathbf{X}[:,n,i], \mathbf{X_i} = \text{SORT}(\mathbf{I_T}, \mathbf{X}), \quad (6)$$

$$T' = T \cdot (1 - P_T), \mathbf{X}' = \mathbf{X_i}\left[: T'\right], \mathbf{X}'' = \mathcal{SN}(\mathbf{X}') \quad (7)$$

Here, $\mathbf{I_T}$ is the SIV calculated for each time step. Based on these scores, $\mathbf{X_i}$ represents the feature maps sorted in descending order of importance. The number of time steps retained, $T'$ is calculated based on a pruning ratio $P_T$. Finally, $\mathbf{X}'$ is the temporal feature after the pruning. Reducing the number of time steps for feature inputs into the spiking neurons and network layers directly reduces the number of neuron states and computational cost, thereby lowering training overhead.

**Computational Overhead Analysis.** For clarity, we choose typical size Spiking-token Mixer models to compare these operations on the ImageNet, as shown in Table 1. The results show that DSTSP significantly lowers the computational overhead. Theoretical analysis is provided in the Appendix A.

Table 1: Comparison of the $OP_{Increase}$ and $OP_{Reduce}$. SMixer-$L$-$D$ denotes for Spiking-token Mixer with $L$ layers and $D$ channels. The Rate represents the ratio of $OP_{Reduce}$ to original cost $OP_{Origin}$.

| Model | $OP_{Increase}(M)\uparrow$ | $OP_{Reduce}(M)\downarrow$ | $OP_{Origin}(M)$ | Rate(%) |
|---|---|---|---|---|
| SMixer-2-256 | 0.33 | 243 | 1106 | 21.97 |
| SMixer-4-384 | 0.45 | 1060 | 4867 | 21.78 |
| SMixer-8-512 | 0.58 | 3741 | 17073 | 21.91 |

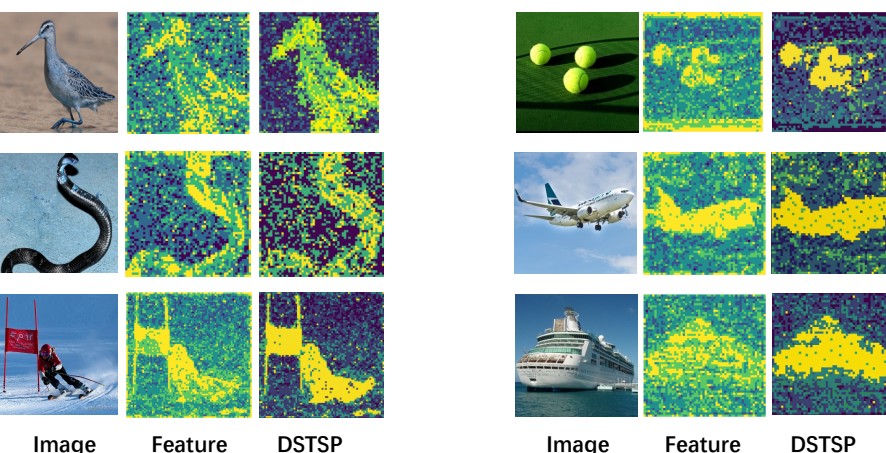

Figure 6: Visualization of SMixer and relating DSTSP on the ImageNet-1K. Feature and DSTSP represent the attention maps generated by the original SMixer module before and after being pruned by our DSTSP method, respectively

## 4 EXPERIMENTS

In this section, we validate DSTSP using Spiking Token Mixer framework as the baseline. We first conduct experiments on classification tasks, including CIFAR10-DVS Li et al. (2017), DVS128 Gesture Amir et al. (2017), CIFAR10, CIFAR100 Krizhevsky (2009), and ImageNet Deng et al. (2009). These datasets cover both neuromorphic and conventional vision domains. Our evaluation centers on training time, energy cost, GPU memory and inference accuracy. We further assess DSTSP on the more demanding task of time series and object detection. Final, the ablation study examines the efficacy, certain strategies, and position of DSTSP. The training epochs, hyperparameter settings are consistent with the original models. Further details are provided in Appendix E.

### 4.1 RESULTS ON IMAGE CLASSIFICATION

**ImageNet.** We evaluate the feasibility of Spiking-token Mixer (SMixer) paradigm Deng et al. (2024) and DSTSP on the ImageNet-1K dataset. We first analyze the effectiveness of DSTSP using the original STMixer-8-512 and STMixer-8-768 as baselines. Besides, we further replace the Spiking Self-Attention in three representative state-of-the-art Spiking Transformer frameworks (SpikformerV2 Zhou et al. (2024b), QKFormer Zhou et al. (2024a), and Spike-driven Transformer-V3 Yao et al. (2025)) with Spiking-token Mixer and further assess DSTSP pruning framework, supported by visual analysis, to demonstrate its efficacy. As detailed in Table 2, replacing SSA with Spiking-token Mixer yields comparable accuracy to the original frameworks at similar parameters, demonstrating its capability to replace the Spiking Transformer and serve as a new prototype. With a fixed spatial pruning ratio of 0.30 and a pruned time step of 1, our DSTSP shows remarkable efficiency gains. Notably, on the STMixer-8-768 model, our pruning method improves accuracy by 0.2 % over the baseline. This high performance are achieved while reducing GPU memory consumption to 76.44% and energy to just 53.03% of the original, alongside a high increase in throughput. When applied to the modified Spikformer-V2 (T=4), our pruning framework results in only a minor 1.3% accuracy loss while also delivering substantial computational savings. Variants based on QK-Former and SDT-v3 also diminish energy consumption and shorten training time while incurring a performance drop of less than 2%. Our qualitative analysis in Figure 6 offers further validation.

Table 2: Performance of DSTSP on ImageNet. Power is calculated as the average theoretical energy consumption when predicting an image from ImageNet test set. The power data for ours is evaluated according to Appendix B and the power data of other works are obtained from related papers. "Model-$L$-$D$" denotes the specific Model with $L$ encoder blocks and $D$ channels. "Model $\rightarrow$ M" indicates the conversion of the original Spikformers Model into the Spiking-token Mixer. We set the spatial pruning ratio to 0.3 and perform temporal pruning to single time step.

| Methods | Architecture | Time Step | Param (M) | TP (im/s) | Memory (MB) | Power (mJ) | Top-1 Acc (%) |
|---|---|---|---|---|---|---|---|
| **STMixer** | STMixer-8-512 | 1 | 30.12 | 211 | 11762 | 2.20 | 73.82 |
| | STMixer-8-768 | 1 | 61.16 | 120 | 17008 | 4.45 | 76.68 |
| **STMixer + DSTSP** | STMixer-8-512 | 1 | 27.61 | 250 | 9578 | 1.68 | 73.56 |
| | STMixer-8-768 | 1 | 58.66 | 162 | 13002 | 2.36 | 76.87 |
| **SpikformerV2** | Spikformer V2-8-384 | 1 | 29.11 | 130 | 5260 | 1.73 | 75.42 |
| | Spikformer V2-8-512 | 1 | 51.55 | 113 | 6880 | 2.84 | 79.05 |
| | Spikformer V2-8-384 | 4 | 29.11 | 82 | 12170 | 4.69 | 78.80 |
| | Spikformer V2-8-512 | 4 | 51.55 | 67 | 18786 | 9.36 | 80.38 |
| **SpikformerV2 $\rightarrow$ M** | Spikformer V2-8-384 | 1 | 27.97 | 156 | 3490 | 1.13 | 76.39 |
| | Spikformer V2-8-512 | 1 | 48.56 | 134 | 5384 | 2.12 | 79.16 |
| | Spikformer V2-8-384 | 4 | 27.97 | 65 | 9256 | 3.65 | 79.12 |
| | Spikformer V2-8-512 | 4 | 48.56 | 56 | 12982 | 7.97 | 80.45 |
| **SpikformerV2 $\rightarrow$ M + DSTSP** | Spikformer V2-8-384 | 1 | 27.34 | 245 | 2584 | 0.83 | 76.22 |
| | Spikformer V2-8-512 | 1 | 47.93 | 198 | 3368 | 1.98 | 78.99 |
| | Spikformer V2-8-384 | 4 | 27.34 | 135 | 7505 | 1.55 | 78.03 |
| | Spikformer V2-8-512 | 4 | 47.93 | 98 | 10387 | 3.44 | 79.15 |
| **QKFormer** | HST-10-384 | 1 | 16.47 | 377 | 13841 | 3.38 | 75.52 |
| | HST-10-512 | 1 | 29.08 | 317 | 20005 | 4.79 | 78.71 |
| **QKFormer $\rightarrow$ M** | HST-10-384 | 1 | 18.31 | 365 | 8385 | 2.76 | 76.03 |
| | HST-10-512 | 1 | 29.70 | 325 | 13051 | 4.06 | 78.69 |
| **QKFormer $\rightarrow$ M + DSTSP** | HST-10-384 | 1 | 14.92 | 461 | 6991 | 2.17 | 75.13 |
| | HST-10-512 | 1 | 25.92 | 398 | 9751 | 3.15 | 77.39 |
| **SDT-V3** | Efficient-transformer-S | 4 | 5.1 | 300 | 9040 | 1.70 | 75.30 |
| | Efficient-transformer-M | 4 | 10.00 | 267 | 12388 | 3.00 | 78.50 |
| | Efficient-transformer-L | 4 | 19.00 | 197 | 16318 | 5.90 | 79.80 |
| **SDT-V3 $\rightarrow$ M** | Efficient-transformer-S | 4 | 6.37 | 373 | 7396 | 1.42 | 75.25 |
| | Efficient-transformer-M | 4 | 10.15 | 280 | 9988 | 2.56 | 78.65 |
| | Efficient-transformer-L | 4 | 19.36 | 219 | 13912 | 4.93 | 79.25 |
| **SDT-V3 $\rightarrow$ M + DSTSP** | Efficient-transformer-S | 4 | 5.05 | 398 | 6060 | 1.01 | 74.15 |
| | Efficient-transformer-M | 4 | 9.42 | 338 | 8210 | 2.02 | 76.63 |
| | Efficient-transformer-L | 4 | 18.66 | 273 | 10710 | 3.91 | 77.75 |

Table 3: Comparision on CIFAR10, CIFAR100, DVS128, CIFAR10-DVS. "Param" denotes "Parameter (M)", "Acc" denotes "Top-1 Accuracy (%)", "$T$" denotes "Time Step".

| Method | CIFAR10 | | | CIFAR100 | | | DVS128 | | | CIFAR10-DVS | | |
|---|---|---|---|---|---|---|---|---|---|---|---|---|
| | Param | $T$ | Acc | Param | $T$ | Acc | Param | $T$ | Acc | Param | $T$ | Acc |
| Spikformer Zhou et al. (2023c) | 9.32 | 4 | 95.51 | 9.32 | 4 | 78.21 | 2.57 | 16 | 98.30 | 2.57 | 16 | 80.90 |
| Spikingformer Zhou et al. (2023a) | 9.32 | 4 | 95.81 | 9.32 | 4 | 78.21 | 2.57 | 16 | 98.30 | 2.57 | 16 | 81.30 |
| CML Zhou et al. (2023b) | 9.32 | 4 | **96.04** | 9.32 | 4 | 80.02 | 2.57 | 16 | 98.60 | 2.57 | 16 | 80.90 |
| S-Transformer Yao et al. (2023a) | 10.28 | 4 | 95.60 | 10.28 | 4 | 78.40 | 2.57 | 16 | **99.30** | 2.57 | 16 | 80.00 |
| **SMixer** | 8.29 | 1 | 95.49 | 8.29 | 1 | 80.00 | 2.63 | 10 | 97.91 | 2.63 | 10 | **83.30** |
| | 8.29 | 4 | 96.01 | 8.29 | 4 | **81.87** | 2.63 | 16 | 98.61 | 2.63 | 16 | 83.02 |
| **SMixer+DSTSP** | **8.24** | 1 | 95.44 | **8.24** | 1 | 79.71 | **2.55** | 10 | 97.56 | **2.55** | 10 | 82.56 |
| | **8.24** | 4 | 95.67 | **8.24** | 4 | 81.03 | **2.55** | 16 | 98.26 | **2.55** | 16 | 82.34 |

The attention maps from Spiking-token Mixer (Feature) effectively highlight key object regions in a manner comparable to the original Spiking Self-Attention. Crucially, we observe that after pruning, the remaining tokens consistently align with the most semantically significant parts of the image (DSTSP). This demonstrates our framework's ability to preserve essential information.

**Cifar and Neuromorphic Datasets.** We then conduct experiments on both static datasets (CIFAR-10, CIFAR-100 Krizhevsky (2009)) and neuromorphic datasets (CIFAR10-DVS Li et al. (2017), DVS128-Gesture Amir et al. (2017)), benchmarking our method against the SMixer baseline as shown in Table 3. By applying a fixed spatial pruning ratio of 0.30 and prune the time-step to 1, we observe a mere 0.5% performance drop on CIFAR-10 and CIFAR-100. On the neuromorphic datasets CIFAR10-DVS and DVS128, our method incurs a performance drop less than 1%.

Table 4: Experimental results of time-series forecasting on 4 benchmarks with various prediction lengths 6, 24, 48, 96. The best results in SNNs are formatted in **bold font format**. ↑ (↓) indicates the higher (lower) the better.

| Model | SNN | Metric | Metr-la | | | | Pems-bay | | | | Solar | | | | Electricity | | | |
|---|---|---|---|---|---|---|---|---|---|---|---|---|---|---|---|---|---|---|
| | | | 6 | 24 | 48 | 96 | 6 | 24 | 48 | 96 | 6 | 24 | 48 | 96 | 6 | 24 | 48 | 96 |
| Transformer | × | $R^2$↑ | .727 | .554 | .413 | .284 | .785 | .734 | .688 | .673 | .953 | .858 | .759 | .718 | .978 | .975 | .972 | .964 |
| | | RSE↓ | .551 | .704 | .808 | .895 | .502 | .558 | .610 | .618 | .223 | .377 | .504 | .545 | .260 | .277 | .347 | .425 |
| Spikformer | ✓ | $R^2$↑ | .697 | .491 | .383 | .242 | .768 | .684 | .678 | .663 | .903 | .819 | .715 | .656 | .956 | .955 | .953 | .943 |
| | | RSE↓ | .581 | .753 | .828 | .917 | .521 | .607 | .613 | .627 | .319 | .439 | .548 | .602 | .371 | .375 | .386 | .450 |
| SMixer | ✓ | $R^2$↑ | **.734** | **.519** | **.422** | **.278** | **.788** | **.726** | **.689** | **.674** | **.945** | **.869** | **.789** | **.732** | .956 | **.971** | **.967** | .952 |
| | | RSE↓ | **.544** | .730 | **.802** | **.895** | **.496** | **.566** | **.603** | .615 | **.216** | **.371** | **.471** | **.531** | .371 | **.305** | **.322** | .389 |
| SMixer + DSTSP | ✓ | $R^2$↑ | .697 | .530 | .400 | .277 | .775 | .712 | .682 | .662 | .930 | .855 | .761 | .662 | **.957** | .965 | .950 | **.955** |
| | | RSE↓ | .581 | **.722** | .816 | .896 | .503 | .577 | .607 | **.596** | .272 | .381 | .490 | .596 | **.369** | .335 | .395 | **.380** |

## 4.2 RESULTS ON TIME SERIES TASK

We evaluate the effectiveness of DSTSP on time-series tasks by conducting experiments on four benchmarks Metr-La, Solar, PEMS-Bay, and Electricity under prediction horizons of 6, 24, 48, and 96 steps. As shown in Table 4, the SMixer paradigm consistently outperforms Spikformer, demonstrating its potential. Besides, integrating DSTSP incurs no significant degradation in performance.

## 4.3 RESULTS ON OBJECT DETECTION TASK

To evaluate the effectiveness of DSTSP on demanding object detection tasks, we first adapt the SpikeYOLO Luo et al. (2024) framework by integrating SMixer blocks, then implement DSTSP. Specifically, we replace the convolutional layers in the third and fourth stages of SpikeY-OLO with 4 and 2 layers of SMixer blocks, respectively. On the challenging MS COCO dataset Lin et al. (2014), our model trained with DSTSP by 20% spatial pruning rate and single pruned timestep and achieved a mAP50 of 57.4, compared to the baseline model's 58.9. We consider this modest decrease in accuracy

Table 5: Performance of detection on COCO val2017 Lin et al. (2014). We benchmark our method against representative baselines of both ANN-to-SNN conversion and direct SNN training paradigms.

| Methods | Architecture | Param (M) | Power (mJ) | Step | mAP@50(%) |
|---|---|---|---|---|---|
| ANN2SNN | Spiking-Yolo [95] | 10.2 | - | 3500 | 25.7 |
| | Spike Calibration [96] | 17.1 | - | 512 | 45.3 |
| | Fast-SNN [13] | 25.1 | - | 15 | 46.4 |
| Direct training | Spiking Retina [97] | 11.3 | - | 4 | 28.5 |
| | EMS-Res-SNN [98] | 26.9 | - | 4 | 50.1 |
| | Meta-SpikeFormer* [28] | 34.9 | 49.5 | 1 | 44.0 |
| | | 75.0 | 140.8 | 1 | 51.2 |
| Direct training | E-SpikeFormer* | 38.7 | 56.2 | 2 | 41.8 |
| | | 38.7 | 94.5 | 4 | 58.4 |
| | | 38.7 | 119.5 | 8 | 58.8 |
| Direct training | SMixer | 49.7 | 36.8 | 4 | **58.9** |
| | SMixer + DSTSP | 39.2 | 21.2 | 4 | 57.4 |

acceptable, given the highly detailed and complex nature of the COCO task. Further deployment details are provided in the Appendix E.3.

## 4.4 ABLATION STUDY

**Spatial-Temporal Pruning Method Ablation.** This section details the implementation of various temporal-spatial pruning configurations on the CIFAR-10 and CIFAR-100 datasets. We evaluate their effects on key metrics, including energy consumption, FLOPs, inference latency, and overall performance. Our findings demonstrate that even at substantial pruning ratios, the proposed strategy preserves model robustness while significantly reducing FLOPs and increasing processing speed. Furthermore, we conduct a granular analysis of the impact of different temporal-spatial pruning ratios. As illustrated in the Appendix E.5, Figure 8 for the DVS-CIFAR10 and CIFAR-100 datasets, a notable degradation is observed only at exceptionally high spatial-temporal pruning levels.

**Comparison of Spatial Pruning Strategies.** To validate the proposed pruning method, a comparative analysis is conducted against two baseline strategies: random pruning (RD) and the pruning of high-SIV tokens (HP). To ensure a fair comparison, the pruning technique is the sole variable, with all other experimental settings held constant. The results affirm our prior findings from the Spatial Temporal Spiking Feature Redundancy analysis, confirming that removing low-SIV regions is the most effective approach for information preservation. As depicted in Figure 7, our method achieve a superior accuracy of 81.03% on the CIFAR100 dataset. In contrast, the HP strategy is the

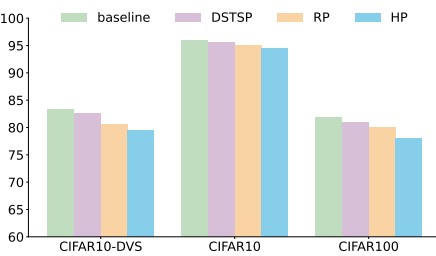

Figure 7: Comparison of Pruning Strategies. The default spatial pruning ratio is set to 0.3, and time step is pruned to 1.

Table 6: Performance of STSFP on CIFAR10 and CIFAR100. We only displayed the GPU, throughput, and FLOPs for CIFAR100 because the results for CIFAR10 are almost the same. We adopt the Softmatch strategy by default to deal with the pruned tokens. TP denotes the throughput.

| $P_s$ | $P_t$ | TP (im/s) | GPU Memory | FLOPs | CIFAR100 | CIFAR10 |
|---|---|---|---|---|---|---|
| 0 | 0 | 266 | 5329M | 3.45G | 81.78 | 96.01 |
| 0.40 | 0 | 337 | 3498M | 2.72G | 80.81 | 95.44 |
| 0.60 | 0 | 368 | 3045M | 2.36G | 79.13 | 94.67 |
| 0 | 0.50 | 671 | 3526M | 2.54G | 81.25 | 95.90 |
| 0 | 0.75 | 667 | 3384M | 2.08G | 81.23 | 95.89 |
| 0.20 | 0.50 | 615 | 2851M | 2.34G | 81.14 | 95.78 |
| 0.30 | 0.75 | 738 | 2346M | 2.04G | 81.03 | 95.67 |

least performant, with its accuracy declining by 2.92% to 78.11%, while the RD strategy yield an intermediate result.

**Comparison with SNN Weight Pruning.** In Table 7, we benchmark DSTSP against conventional SNN pruning methods. A direct one-to-one comparison is non-trivial, as DSTSP's methodology diverges significantly from standard synaptic pruning. Consequently, we evaluate performance based on three key criteria: the pruning cost, the degree of sparsity achieved, and the accuracy degradation relative to the original dense model. By integrating pruning directly into the training loop for efficiency, DSTSP's computational cost is treated as a single, unified stage. Notably, our proposed method incurs the minimal performance loss among all techniques benchmarked.

**Position of DSTSP.** This section analyzes the optimal placement of the pruning module by considering its temporal and spatial operations. A critical constraint dictates that spatial pruning must be performed subsequent to the SPS module, as its 2D convolutional layers depend on the original feature map dimensions (Height × Width) that spatial pruning would otherwise alter. The empirical results for different module placements are presented in Table 8. The findings indicate that applying DSTSP during both the training and inference phases (B) yields the highest performance, though employing it exclusively during training also produces commendable results. We further explore strategies involving pruning only during training (T) or inference (I). The performance of the training-only approach is comparable to that of using DSTSP in both stages, whereas applying it solely during inference results in inferior performance.

Table 7: We define the pruning cost as the ratio $\frac{N_p}{N_d}$ , where $N_p$ is the number of epochs for pruning and $N_d$ is for training the original dense model. Since our method integrates pruning into the standard training process without requiring additional epochs, its cost is 1. Furthermore, we note that our reported sparsity measures the ratio of pruned spiking tokens, whereas in prior works it typically refers to the ratio of zero-valued weights.

| Pruning Method | Base Acc. (%) | Cost | Pruned Acc. (%) | Sparsity |
|---|---|---|---|---|
| NDSNN Huang et al. (2023) | 69.86 | 5.00 | 68.07 | 0.90 |
| | | | 66.73 | 0.95 |
| | | | 63.51 | 0.98 |
| Grad R Chen et al. (2021a) | 71.34 | 34.13 | 67.47 | 0.95 |
| | | | 67.31 | 0.98 |
| IMP Chen et al. (2021a) | 71.34 | 37.67 | 71.38 | 0.90 |
| | | 56.33 | 70.54 | 0.96 |
| | | 70.33 | 67.35 | 0.98 |
| RCMO-SNN Chen et al. (2023) | 74.71 | 51.67 | 72.67 | 0.95 |
| | | 65.67 | 70.80 | 0.97 |
| DSTSP | 81.87 | **1** | 80.67 | 0.50 |
| | | | 80.13 | 0.60 |
| | | | 79.78 | 0.70 |

Table 8: Position of STSFP. For the 'Stage', B indicates that STSFP is used during both training and inference, T indicates usage during training only, and I indicates usage during inference only. Regarding placement, IS denotes integrating the TSFP within the SPS, while AS indicates positioning the module after the SPS. For CIFAR10-DVS, $P_s = 0.3$ and $P_t = 0.9$. For CIFAR100, $P_s = 0.3$ and $P_t = 0.75$.

| Dataset | Stage | TSFP | SSTP | Acc. |
|---|---|---|---|---|
| CIFAR10-DVS | B | AS | AS | 82.56 |
| | B | IS | AS | 82.03 |
| | T | AS | AS | 82.45 |
| | I | AS | AS | 80.05 |
| CIFAR100 | B | AS | AS | 81.03 |
| | B | IS | AS | 80.56 |
| | T | AS | AS | 80.95 |
| | I | AS | AS | 79.74 |

## 5    CONCLUSION

We begin by analyzing three major challenges currently faced by deep SNNs: event-driven constraints, performance limitations, and training overhead. Specifically, we explore the potential of the Spiking-token Mixer as a prototype in terms of both performance and event-driven characteristics. By integrating SMixer into mainstream SNN variants, we demonstrate that its performance is on par with prior Spiking Transformers, highlighting the potential for high-performance architectures that are fully event-driven. To address the excessive training cost associated with SMixer, we propose Dynamic Spatial-Temporal Spiking Pruning, which reduces training overhead by pruning redundant spiking features while maintaining competitive performance. We hope that SMixer can inspire future research in the development of deep SNNs.

## 6    ACKNOWLEDGMENTS AND DISCLOSURE OF FUNDING

We sincerely thank Dr. Shikuang Deng for his assistance with this work. This work was supported by CAS Project for Young Scientists in Basic Research (YSBR-116), National Natural Science Foundation of China (62325603, 62236009, U22A20103), Beijing Science and Technology Plan (Z241100004224011), the 14th Five-Year Development Plan (2021-2025) Research Grant of National Education Examinations Authority, China, and Qingdao Municipal Science and Technology Benefit People Demonstration and Guidance Special Project under Grant 21-1-4-sf-2-nsh.

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

## A    COMPUTATIONAL OVERHEAD ANALYSIS

Although they do not introduce additional training parameters, the inclusion of DSTSP in Spiking-token Mixer incurs some computational overhead. For a spiking feature $\mathbf{X} \in \mathbb{S}^{T \times N \times C}$, the total number of additional operations brought by DSTSP is:

$$\text{OP}_{\text{DSTSP}} = (2 - P_{\text{t}}) \cdot TNC + ((1 - P_{\text{t}}) \cdot T + 1)N^2 + T^2,$$

where $P_{\text{t}}$ denotes temporal pruning ratio and $P_{\text{s}}$ denotes spatial pruning ratio. Here we choose the Bubble Sort when discussing computational overhead, using it as the upper bound for the cost of DSTSP. Specifically, for DTSP, the number of operations required for SIV addition is $TNC$, while the number of operations required for sorting is $T^2$ (including comparisons and swaps). For the DSSP applied to the features $\mathbf{X}_{\text{tp}} \in \mathbb{S}^{T_{\text{p}} \times N \times C}, T_{\text{p}} = T \cdot (1 - P_{\text{t}})$ after DTSP, the number of operations required for SIV addition is $(1 - P_{\text{t}}) TNC$, while the number of operations required for sorting is $(1 - P_{\text{t}}) TN^2$. In addition, the indexing operation on the attention weights requires an additional computational cost of $N^2$.

The reduced number of operations for which the $L$-layers network inference overhead after pruning is:

$$\text{OP}_{\text{Reduce}} = L \cdot P_{\text{t}} \cdot T \cdot (P_{\text{s}} \cdot 10NC^2 + P_{\text{s}}^2 \cdot N^2 C). \tag{8}$$

Besides, the total number of operations introduced by the original Spiking-token Mixer is:

$$\text{OP}_{\text{Origin}} = L \cdot T \cdot (10NC^2 + N^2 C). \tag{9}$$

The preceding equations lead to the conclusion that for large values of $P_s$ and $P_t$, the $\text{OP}_{\text{reduce}}$ with our proposed DSTSP is considerably lower than that of the original Spiking-token Mixer $\text{OP}_{\text{Origin}}$. Furthermore, the computational overhead introduced by the DSTSP operation, $\text{OP}_{\text{DSTSP}}$ is negligible. During training, the parameter-free DSTSP does not require training or gradient computation. Additionally, pruned features do not need gradient back-propagation, further accelerating the training process.

## B    THEORETICAL ENERGY CONSUMPTION

According to the general convention of SNNs ( Panda et al. (2020); Yao et al. (2023b)), we posit that the MAC and AC operations are executed on 45nm hardware (Horowitz, 2014), with energy consumption values of $EC_{\text{MAC}} = 4.6\text{pJ}$ and $EC_{\text{AC}} = 0.9\text{pJ}$ per operation, respectively. The theoretical Energy Consumption (EC) of ANNs can be derived as follows:

$$EC_{\text{ANN}} = 4.6\text{pJ} \times \text{MACs}. \tag{10}$$

In SNNs, the AC operations can be obtained by multiplying the MAC operations by the firing rate $f$ of input spikes and the simulation time step $T$,

$$\text{ACs} = \text{MACs} \times f \times T. \tag{11}$$

The operations of the first layer are MACs to map the floating-point original image to spike features, while subsequent-layers operations are ACs for modeling sparse spiking features,

$$EC_{\text{SP}} = 4.6\text{pJ} \times \text{MACs}^1 + 0.9\text{pJ} \times \sum_{l=2}^{L} \text{ACs}^l, \tag{12}$$

where $L$ denotes the number of linear layers in the models. Note that we ignore the energy of BN, as it can be incorporated into the linear layers during inference.

## C    THEORETICAL ANALYSIS OF SPATIAL-TEMPORAL SPIKING FEATURE REDUNDANCY

We propose two metrics to support quantitative analysis in Sec 3.2.

**Ratio of Information Entropy (RIE)** is employed to quantify the similarity of information content between two feature sets, with a score approaching 1 indicating a higher degree of resemblance. The RIE is formally defined as the ratio of the information entropy of the pruned feature map $IE(\mathbf{X}_p)$, to that of the original feature map $IE(\mathbf{X})$. The calculation for information entropy is given by:

$$IE(\mathbf{X}) = - \sum_{i=0,1} P_{\mathbf{X}=i} \log_2 P_{\mathbf{X}=i}, \tag{13}$$

and the RIE is subsequently computed as:

$$RIE(\mathbf{X}p, \mathbf{X}) = \frac{IE(\mathbf{Xp})}{IE(\mathbf{X})}, \tag{14}$$

where $P_{\mathbf{X}=i}$ represents the probability of a given spike value (0 or 1). In our implementation, this probability term is empirically estimated by measuring the mean spike firing frequency across all samples in the test set.

**Multi-Scale Structural Similarity (MS-SSIM)** quantifies the structural congruence between two images across multiple scales. An MS-SSIM score approaching its maximum value of 1 signifies a higher degree of similarity. We employ this metric, in conjunction with feature visualization techniques, to analyze features in both the spatial and temporal domains.

## D  THE DSTSP ALGORITHM WORKFLOW

The complete algorithmic details of DSTSP are presented below in Algorithm 1.

---

**Algorithm 1** Dynamic Spatial-Temporal Spiking Pruning

---

**Require:** Spiking Feature $\mathbf{X} \in \mathbb{S}^{T \times N \times C}$, Spatial Pruning Ratio $P_s$, Temporal Pruning Ratio $P_t$
**Ensure:** Pruned Spiking Feature $\mathbf{X}_p \in \mathbb{S}^{T_p \times N_p \times C}$, where $T_p = T \cdot (1 - P_t)$ and $N_p = N \cdot (1 - P_s)$
 1: **Dynamic Temporal Spiking Pruning**
 2: Sum the $\mathbf{X}$ for each time step to obtain $\mathbf{SIV}_t$
 3: Sort $\mathbf{SIV}_t$ to obtain indices of the top $T_p$ largest correspondences in $\mathbf{X}$
 4: Select $\mathbf{X}_{tp}$ according to the indices for Dynamic Spiking Spatial Pruning
 5: **Dynamic Spatial Spiking Pruning**
 6: **for** $j = 1$ to $T_p$ **do**
 7:     Sum spikes of each token in the $\mathbf{X}_j$ to obtain $\mathbf{SIV}_s$
 8:     Sort $\mathbf{SIV}_s$ to obtain indices of the top $N_p$ largest correspondences in $\mathbf{X}_j$
 9:     Select $\mathbf{X}_{j,\text{sp}}$ according to the indices
10:     Select $\mathbf{W}_{M,\text{sp}}$ according to the indices
11: **end for**

---

## E  EXPERIMENTAL DETAILS

### E.1  IMAGENET-1K EXPERIMENTAL SETTINGS

ImageNet-1K dataset Deng et al. (2009) is commonly used for computer vision tasks. It spans 1000 object classes and contains around 1.3 million training images and 50,000 validation images. For experiments on the ImageNet dataset, we applied the DSTSP methodology to STMixer and three Spikformers variants SpikformerV2, QKFormer, and SDTV3 by replacing their original token-mixing operators with the corresponding Spiking-token Mixer. We keep all training hyper-parameters identical to those reported in their respective papers and conduct training on 8 NVIDIA-4090 GPUs.

### E.2  SMALL DATASETS EXPERIMENTAL SETTINGS

All experiments on CIFAR10 and CIFAR100 are conducted on four 3090 GPUs. We employ the same training script as SMixer and employed identical data augmentation techniques. Additionally, we conduct further experiments on two neuromorphic datasets to demonstrate the effectiveness of

DSTSP. CIFAR10-DVS Li et al. (2017) is a neuromorphic dataset that is obtained from the CIFAR-10 dataset through a DVS camera. There are 10k images in CIFAR10-DVS, and we split them into 9k training images and 1k test images. We follow STMixer and downsample the image resolution from 128×128 to 48×48. DVS128 Gesture Amir et al. (2017) is a gesture recognition dataset that consists of 11 hand gesture classes performed by 29 individuals under 3 different lighting conditions. Notably, we use the ILIF neurons Luo et al. (2024) in all our experiments.

### E.3 COCO EXPERIMENTAL SETTINGS

The prevalent reliance of existing SNN object detection models on convolutional networks makes it a non-trivial task to create a framework based on a Spiking-token mixer with DSTSP. Notably, even recent architectures such as the Meta Spike-driven Transformer V2 Yao et al. (2024), SpikeY-OLO Luo et al. (2024) are still fundamentally structured around a large number of convolutional layers. To demonstrate the capability of the Spiking-token Mixer and our DSTSP training methodology for challenging object detection tasks, we introduce a hybrid architecture that circumvents the heavy dependence on convolution. Our approach adapts the advanced SpikeYOLO model by integrating Spiking-token Mixer blocks at critical feature extraction stages. Specifically, we replace the convolutions in the third and fourth stages of SpikeYOLO with 4 layers and 2 layers of Spiking-token Mixer blocks, respectively. During training, we fix the input resolution at 640×640. In the inference, though we rescale the image while preserving its original aspect ratio and set the longer side to a fixed maximum of 672 pixels, their spatial dimensions remain non-uniform. Before entering each SMixer block, we first convert the feature map to 640×640 via adaptive average pooling; after the block, we bilinearly interpolate the output back to the original resolution. However, such resolution switching still incurs non-negligible accuracy degradation. Thus, future work will aim to develop more advanced feature-pruning algorithms for similarly complex visual tasks.

### E.4 TIME SERIES TASK EXPERIMENTAL SETTINGS AND RESULTS

**Datasets** The key statistics and data distribution for each dataset are summarized in the Table 9.

Table 9: The statistics of time-series datasets.

| Dataset | Samples | Variables | Observation Length | Train-Valid-Test Ratio |
|---------|---------|-----------|--------------------|-----------------------|
| Metr-la | 34,272 | 207 | 12, (short-term) | (0.7, 0.2, 0.1) |
| Pems-bay | 52,116 | 325 | 12, (short-term) | (0.7, 0.2, 0.1) |
| Solar-energy | 52,560 | 137 | 168, (long-term) | (0.6, 0.2, 0.2) |
| Electricity | 26,304 | 321 | 168, (long-term) | (0.6, 0.2, 0.2) |

**Metrices** For evaluating time-series forecasting performance, we employ two primary metrics: the coefficient of determination ($R^2$) and the Root Relative Squared Error (RSE).

$$R^2 = 1 - \frac{\sum_{m=1}^{M} \sum_{c=1}^{C} \sum_{l=1}^{L} (Y_{c,l}^m - \hat{Y}_{c,l}^m)^2}{\sum_{m=1}^{M} \sum_{c=1}^{C} \sum_{l=1}^{L} (Y_{c,l}^m - \bar{Y}_{c,l})^2}, \tag{15}$$

$$\text{RSE} = \frac{\sqrt{\sum_{m=1}^{M} ||\mathbf{Y}^m - \hat{\mathbf{Y}}^m||_2}}{\sqrt{\sum_{m=1}^{M} ||\mathbf{Y}^m - \bar{\mathbf{Y}}||_2}}. \tag{16}$$

In the above Formulas ( 15), the variable $M$ indicates the total number of test samples, $C$ specifies the channel count, and $L$ defines the prediction horizon. The term $\bar{Y}$ is the average value of $\mathbf{Y}^m$. A specific notation, $Y_{c,l}^m$, denotes the value at the $l$-th future time step for the $c$-th variable within the $m$-th sample. Concurrently, $\bar{Y}_{c,l}$ represents the mean of $Y_{c,l}^m$ computed over all samples. Ground truth values are represented by the symbols $\mathbf{Y}^m$ and $Y_{c,l}^m$. These metrics are chosen over alternatives like Mean Squared Error (MSE) or Mean Absolute Error (MAE) due to their enhanced robustness to the scale of dataset values, rendering them highly suitable for the time-series forecasting context.

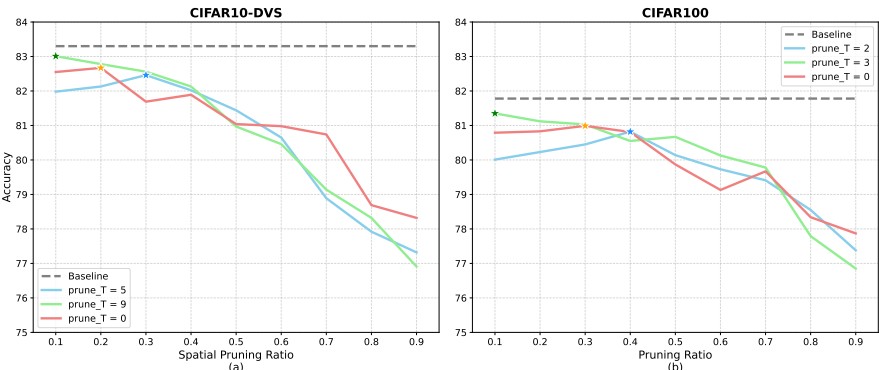

Figure 8: Performance of Spatial-Temporal Spiking Feature Pruning on CIFAR10-DVS and CI-FAR100. The temporal pruning ratio is shown in the figure, `prune_T = n` means that we discard n time steps. The original time step in CIFAR10-DVS is set to 10, and CIFAR100 is 4. The spatial pruning ratio is varied from 0.1 to 0.9.

**Model Architecture and Training Hyper-parameters** A fixed setting of 4 time steps is used for all SNNs. We build a model inspired by the working CPG Lv et al. (2025), replacing Spiking Self-Attention with the SMixer architecture, and introduce the DSTSP pruning mechanism before the first SMixer block. All variants are built with 2 blocks. The dimension for features is set to 256, while the hidden feature dimension within the FFN is 1024. We configure the training process with a batch size of 64 and utilized the Adam optimizer. The learning rate was managed by a cosine scheduler, starting at $1 \times 10^{-4}$. To prevent overfitting, an early stopping mechanism was implemented with a patience of 30 epochs. All experiments are executed on 32G-V100 GPUs.

### E.5 MORE ABLATION RESULTS ON SPATIAL-TEMPORAL PRUNING RATIO

As shown in Figure. 8, we present the performance of our model on the CIFAR10-DVS and CI-FAR100 datasets under various spatial-temporal pruning ratios. We observe that under a high temporal pruning ratio, performance degrades as the spatial pruning ratio increases. This suggests that the model requires sufficient information to make accurate inferences. Conversely, with a low temporal pruning ratio, we find that a certain degree of spatial pruning can lead to improved performance. For instance, on the CIFAR100 dataset, the best performance is achieved with a spatial pruning ratio of 0.4 when half of the time steps are pruned (i.e., two remaining time steps). Furthermore, when both spatial and temporal pruning ratios are moderate, the performance loss remains marginal, which demonstrates the robustness of our proposed pruning method.

## F USE OF LLMS

We declare that the LLMs are used solely to aid or polish the writing and are not involved in the development of the main methodology or comparative experiments.

