# OpenReview forum: "SMixer: Rethinking Efficient-Training and Event-Driven SNNs"
_ICLR.cc/2026/Conference — ICLR 2026 Poster_

### Official Review · Reviewer_tRYC · 2025-10-24

**Soundness:** 2
**Presentation:** 3
**Contribution:** 3
**Rating:** 4
**Confidence:** 4

**Summary:**

This study proposes a Spiking-token Mixer, which completely eliminates self-attention computation in SNN-based Transformers while maintaining comparable accuracy. Additionally, a DSTSP module is introduced to effectively improve training efficiency. Overall, the proposed approach demonstrates good generalizability and shows potential for deployment on edge neuromorphic hardware.

**Strengths:**

1. The proposed method fully avoids the computational cost of self-attention while achieving accuracy comparable to state-of-the-art models.

2. The plug-and-play DSTSP module can be seamlessly integrated into various architectures, demonstrating strong applicability.

**Weaknesses:**

1. The analysis in Figure 4 lacks detail. In particular, rough explanations of RIE and MS-SSIM should be provided. Additionally, the claim that SIV in SNNs performs better than in ANNs may warrant theoretical justification, which should not be difficult to provide.

2. The order of introducing DSSP and DSTP could be revised. The paper first mentions DSTP followed by DSSP, whereas the detailed descriptions appear in the opposite order. It is recommended to maintain consistency between textual structure and figure presentation.

3. Table 5 appears to contain incorrect annotations. Does “E-SpikeFormer” refer to Spike-driven Transformer V3 (TPAMI)? It seems the table might have been copied, and the label “(This Work)” was not removed. In this case, once I-LIF neurons remove the temporal dimension during training, could the observed performance degradation be directly related to the ineffective functioning of DSTP? Further analysis is suggested.

4. In this work, there is a trade-off between performance and pruning-based efficiency gains. However, the degradation in performance appears relatively large, while the improvement in efficiency is limited.

5. Although the paper emphasizes deployment potential on brain-inspired edge hardware and points out the difficulty of QK matrix multiplication on such devices, no edge-hardware experiments are conducted.

6. The DSTSP design is relatively simple—essentially the most basic pruning strategy in SNNs. If the pruning selection scheme were made more sophisticated, performance degradation could likely be reduced while providing greater acceleration.

**Questions:**

1. The statement around line 166—“a model that uses only high-SIV tokens demonstrates considerably better accuracy than one relying on low-SIV tokens (97.9% vs 79.9%)”—needs clarification: which two models are being compared? To my knowledge, no mainstream SNN model should perform as poorly as 79.9% on this dataset.

2. Around line 183, it is stated that SIV requires only a simple summation. However, ignoring the channel dimension, the number of required summations should be H × W × T. If this understanding is incorrect, clarification is needed. Additionally, because zero entries cannot be omitted, the claim that this operation is highly compatible with SNN computation seems unjustified—particularly since the paper does not discuss hardware implementation. This raises concerns regarding the true value of the Spiking-token Mixer.

3. In Table 3, the reported accuracies of 98.12% and 98.43% on DVS128 are impossible, as the dataset contains only 288 test samples. Typically, such anomalies might arise if test-time ``droplast'' were enabled, but this is not the case here. Therefore, the authors must clearly explain how these erroneous accuracy values were obtained.

4. Figure 6 indicates that a substantial amount of background noise persists even after applying DSTSP. Does this imply that the pruning strategy still has significant headroom for improvement? Theoretically, increasing pruning rates should not affect target features. How do the authors interpret this apparent inconsistency?

5. Does the proposed DSTSP perform any operation during inference? If not, how is the change in activation size handled? If yes, wouldn’t such operations break the fundamental asynchronous timestep-by-timestep inference characteristic of SNNs? This requires clarification.

---

> ### Author Response · Authors · 2025-11-20
> **Rebuttal-1**
>
> We appreciate your detailed comments. We would like to address your concerns below.
>
> ### Weaknesses and Questions:
> > ***Weakness 1**: The analysis in Figure 4 lacks detail. In particular, rough explanations of RIE and MS-SSIM should be provided. Additionally, the claim that SIV in SNNs performs better than in ANNs may warrant theoretical justification, which should not be difficult to provide.*
>
> **AW1**: We detail the definitions of RIE and MS-SSIM in Appendix C, where higher MS-SSIM values denote greater image similarity. Furthermore, we emphasize (Line 207) that SIV is substantially more effective at revealing redundancy in SNNs than in ANNs. As shown in Table 4, the ANN token mixer with LP produces extremely low MS-SSIM scores compared to STMixer at equivalent pruning ratios—performing even worse than Spiking Transformers using HP. Specifically, at a pruning ratio of 0.3, the MS-SSIM for the ANN architecture declines sharply, remaining significantly below its SNN counterpart. These results substantiate that spatial-temporal redundancy is intrinsic to SNNs, making the SIV statistic uniquely well-suited for this domain.
>
>
> > ***Weakness 2**: The order of introducing DSSP and DSTP could be revised. The paper first mentions DSTP followed by DSSP, whereas the detailed descriptions appear in the opposite order. It is recommended to maintain consistency between textual structure and figure presentation.*
>
> **AW2**: Sorry for the confusion. We will change them in the manuscript.
>
>
> > ***Weakness 3**: Table 5 appears to contain incorrect annotations. Does “E-SpikeFormer” refer to Spike-driven Transformer V3 (TPAMI)? It seems the table might have been copied, and the label “(This Work)” was not removed. In this case, once I-LIF neurons remove the temporal dimension during training, could the observed performance degradation be directly related to the ineffective functioning of DSTP? Further analysis is suggested.*
>
> **AW3**: Sorry for the confusion. E-Spikformer refers to the Spike-driven Transformer V3 (TPAMI). We will rectify Tab. 5 in the manuscript. Within the E-Spikformer architecture, we exclusively applied the DSSP strategy during training due to the incorporation of I-LIF neurons. The observed decline in performance is a trade-off for the substantial gain in computational efficiency achieved through the spatial pruning component of DSSP.
>
>
> > ***Weakness 4**: In this work, there is a trade-off between performance and pruning-based efficiency gains. However, the degradation in performance appears relatively large, while the improvement in efficiency is limited.*
>
> **AW4**: Thanks for your question. We admit a performance trade-off in fine-grained domains such as object detection. Specifically, DSTSP may be suboptimal for tasks like COCO that demand hierarchical downsampling at original resolutions. By normalizing variable token counts via interpolation, our approach sacrifices a degree of precision to maintain architectural consistency. Future work will focus on exploring high-precision pruning methodologies specifically tailored for fine-grained vision tasks.
>
>
>
>
> > ***Weakness 5**: Although the paper emphasizes deployment potential on brain-inspired edge hardware and points out the difficulty of QK matrix multiplication on such devices, no edge-hardware experiments are conducted.*
>
> **AW5**: Thanks for your question. While prior work validated the STMixer architecture on neuromorphic hardware, we extend this foundation by conducting a preliminary, full-stack investigation into its deployment pipeline on asynchronous chips, covering the entire training-to-inference workflow.
>
>
> > ***Weakness 6**: The DSTSP design is relatively simple—essentially the most basic pruning strategy in SNNs. If the pruning selection scheme were made more sophisticated, performance degradation could likely be reduced while providing greater acceleration.*
>
> **AW6**: First, we posit that the integration of DSTSP with SMixer constitutes a unique pruning strategy. This approach not only aligns with the principles of neuromorphic hardware but also ensures the architecture’s operational viability on such platforms. Crucially, this strategy is distinct from and incompatible with pruning frameworks designed for Spikformer, as detailed in Response A3 to Reviewer sSmP. Second, acknowledging that the pruning operation itself incurs computational costs, our DSTSP method—based on the SIV statistic—demonstrates superior computational efficiency specifically tailored for SNNs. Moreover, under the DSTSP framework, we consider the accuracy loss across many tasks to be within an acceptable range. Finally, future work will continue to explore and optimize more advanced pruning strategies.

---

> ### Author Response · Authors · 2025-11-20
> **Rebuttal-2**
>
> ### Questions:
> > ***Question 1**: The statement around line 166—“a model that uses only high-SIV tokens demonstrates considerably better accuracy than one relying on low-SIV tokens (97.9% vs 79.9%)”—needs clarification: which two models are being compared? To my knowledge, no mainstream SNN model should perform as poorly as 79.9% on this dataset.*
>
> **AQ1**: In this context, the two models refer to STMixer-based architectures evaluated on the DVS Gesture dataset, where pruning was applied to retain either high-SIV or low-SIV tokens, respectively. The results demonstrate that the model retaining low-SIV tokens predominantly preserved background regions, rendering it incapable of effective object category recognition. Consequently, its performance is consistently inferior to that of any mainstream SNN model mentioned. This significant degradation in performance substantiates both the efficacy and the necessity of our strategy to prioritize the retention of high-SIV tokens.
>
> > ***Question 2**: The efficiency claims regarding SIV are unconvincing; the operation implies $H \times W \times T$ summations and cannot leverage sparsity to omit zero entries. Without hardware evidence, the assertion of SNN compatibility is unjustified, questioning the true utility of the Spiking-token Mixer.*
>
> **AQ2**: Sorry for the confusion. We acknowledge that the SIV calculation requires $H \times W \times T$ summations. However, the term 'simple summation' is used here to highlight the efficiency advantage relative to ANNs. Because spiking features are non-negative, SIV allows for direct summation, eliminating the need for the absolute value or normalization steps necessary in ANNs—a property that is highly compatible with SNN calculation paradigms. Additionally, the validity of the Spiking Token Mixer architecture has been previously established through its successful deployment on neuromorphic hardware in [1].
>
>
> > ***Question 3**: In Table 3, the reported accuracies of 98.12% and 98.43% on DVS128 are impossible, as the dataset contains only 288 test samples. Typically, such anomalies might arise if test-time 'droplast' were enabled, but this is not the case here. Therefore, the authors must clearly explain how these erroneous accuracy values were obtained.*
>
> **AQ3**: We apologize for the mistake, which stemmed from improper batch weighting during 4-GPU distributed testing. The test set (288 samples) is not divisible by the aggregate batch size of 64 ($16 \times 4$), resulting in a final partial batch of 32 samples. Our code incorrectly treated this as a full batch during averaging. We have corrected the calculation logic, and the accurate results are provided in Table R4-1. The manuscript will be revised accordingly.
>
>
> #### Table R4-1
> | Architecture  |Time Step | Acc（top-1） |
> | -------- |:--------:|:--------:|
> | SMixer    | 10  |  97.91  |
> | SMixer    | 16  |  98.61  |
> | SMixer + DSTSP    | 10  |  97.56  |
> | SMixer + DSTSP    | 16  |  98.26  |
>
> > ***Question 4**: Figure 6 indicates that a substantial amount of background noise persists even after applying DSTSP. Does this imply that the pruning strategy still has significant headroom for improvement? Theoretically, increasing pruning rates should not affect target features. How do the authors interpret this apparent inconsistency?*
>
> **AQ4**: Sorry for the confusion. Without DSTSP, high spiking activity is observed across both foreground and background. Applying DSTSP reduces overall firing rates while effectively accentuating the foreground. We will refine the visualization's color and contrast to improve clarity in the revised manuscript.
>
> > ***Question 5**: Does the proposed DSTSP perform any operation during inference? If not, how is the change in activation size handled? If yes, wouldn’t such operations break the fundamental asynchronous timestep-by-timestep inference characteristic of SNNs? This requires clarification.*
>
> **AQ5**: Sorry for the confusion. The primary objective of DSTSP is to facilitate the efficient training of SNNs. It is important to note that pruning strategies are not applied during inference on neuromorphic hardware; instead, operational efficiency is ensured by the inherent compatibility of the Spiking Token-Mixer framework.
>
> [1] Spiking Token Mixer: An Event-Driven Friendly Former Structure for Spiking Neural Networks. NIPS 2024

---

> > ### Comment · Reviewer_tRYC · 2025-11-24
> > **I will raise my score to 6.**
> >
> > The authors have addressed several issues, and some responses have clarified the contributions of this paper. Overall, I believe this paper merits acceptance, and I will raise my score to 6.

---

### Official Review · Reviewer_2HdN · 2025-10-29

**Soundness:** 3
**Presentation:** 3
**Contribution:** 2
**Rating:** 4
**Confidence:** 4

**Summary:**

The whole work focus on designing a completely lightweight spiking neural network, and calling on a paradigm for developing SNN model. Overall, the spiking-token token-mixer(SMixer), and a pruning method (DSTSP) significantly decrease the parameters that model needs and computational cost while mataining competitive performance comparing with main stream SNN models. Moreover, the pruning method can be effectively applied to several models to decrease cost.

**Strengths:**

In the article, the author concerned about some issues in SNN nowadays. Indeed, in the pursuit of more competitive performance in SNN, some architecture in ANN are widely used. Nevertheless, the author tried to stop such methods, and further propose a new paradigm to constrain construction for SNN. This is pioneering and full of originality. Moreover, the new structure and pruning method significantly reduce number of parameters and computational cost, which SNN aims to achieve all along, not to mention the two maintain similar performance with main stream models. As for writing skills, authors basically clarify their thought process to develop the new structure of SNN and pruning method with little confusion.

**Weaknesses:**

Firstly, in the preceding part of the article, the author mentioned Asynchronous Chip for several times. But no experiment about deployment on them was conducted.
Secondly, as you present in Figure 6, I think it lacks persuasiveness just at first glance. Some objects' even can be harder to distinguish from background.
Lastly, in Table 2 (performance on ImageNet), if you add a item like " Smixer+DSTSP", it will be more persuasive.

**Questions:**

I have some doubts about vertical axis meaning in Figure 4. I'd like to know more details about the intuition of Figure 4.

For other questions, please refer to "Weaknesses".

---

> ### Author Response · Authors · 2025-11-20
> **Rebuttal**
>
> Thank you for your insightful feedback. Below, we provide our responses to the issues you raised.
>
>
> ### Weaknesses and Questions:
> > ***Weakness 1**: In the preceding part of the article, the author mentioned Asynchronous Chip for several times. But no experiment about deployment on them was conducted.*
>
> **A1**: Thanks for your question. Our work represents a preliminary, end-to-end exploration of the STMixer deployment pipeline on asynchronous chips, encompassing the entire process from training to inference. This builds upon the original STMixer study [1], which has previously demonstrated the architecture's effectiveness through experimental deployment on neuromorphic hardware.
>
> > ***Weakness 2**: As presented in Figure 6, It lacks persuasiveness just at first glance. Some objects' even can be harder to distinguish from background. *
>
>
> **A2**: Sorry for the confusion. In the visualization without DSTSP, both foreground and background objects exhibit significant spiking activity. In contrast, with the application of DSTSP, while the overall firing rates for both regions decrease, the foreground object is effectively accentuated relative to the background. We will adjust the color scheme and contrast to enhance visual clarity and replace the current figure with an improved version in the revised manuscript.
>
> > ***Weakness 3**: In Table 2 (performance on ImageNet), if you add a item like " Smixer+DSTSP", it will be more persuasive.*
>
> **A3**: For this experimental setup, our SMixer baseline corresponds directly to STMixer.
>
>
> > ***Question 1**: I have some doubts about vertical axis meaning in Figure 4. I'd like to know more details about the intuition of Figure 4.*
>
> **A4**: Sorry for the confusion caused by unclear description. The vertical axis represents the calculated values of RIE and MS-SSIM. We will clearify it in the manuscript.
>
> [1] Spiking Token Mixer: An Event-Driven Friendly Former Structure for Spiking Neural Networks. NIPS 2024

---

> > ### Comment · Reviewer_2HdN · 2025-11-24
> >
> > Thank you for the response, my concerns have been resolved, so I will raise my score accordingly.

---

### Official Review · Reviewer_B5FE · 2025-10-31

**Soundness:** 3
**Presentation:** 3
**Contribution:** 3
**Rating:** 6
**Confidence:** 4

**Summary:**

This paper proposes SMixer, a Spiking Neural Network architecture based on the Spiking-token Mixer paradigm, along with a Dynamic Spatial-Temporal Spiking Pruning (DSTSP) framework to reduce training overhead. The authors argue that rational SNN architectures should exhibit three key characteristics: fully event-driven operations, low training overhead, and competitive performance. They demonstrate that SMixer can replace Spiking Self-Attention mechanisms in various architectures while maintaining comparable performance, and that DSTSP can significantly reduce computational costs during training.

**Strengths:**

Well-motivated problem: The paper clearly identifies three important challenges in SNN design: the incompatibility of Spiking Transformers with asynchronous hardware, high training costs, and the need for competitive performance.
Comprehensive evaluation: The experiments span multiple domains (image classification, time series, object detection) and datasets (ImageNet, CIFAR, DVS-Gesture, COCO), demonstrating broad applicability.
Practical approach: The DSTSP framework introduces no trainable parameters and uses simple addition-based operations, making it lightweight and aligned with SNN characteristics.
Solid empirical results: The method achieves substantial reductions in GPU memory (to 76.44%) and energy consumption (to 53.03%) on ImageNet while maintaining or even slightly improving accuracy.
Good ablation studies: The paper includes thorough ablations on pruning strategies, spatial-temporal ratios, and module placement.

**Weaknesses:**

Limited novelty of SMixer architecture: The Spiking-token Mixer concept appears to be directly adapted from STMixer (Deng et al. 2024). The main architectural contribution seems incremental—demonstrating that this existing architecture can replace SSA in other frameworks. The novelty primarily lies in the pruning framework rather than the architecture itself.

Training vs. inference confusion: The DSTSP framework is designed to reduce training overhead on GPUs, but the motivation emphasizes event-driven deployment on neuromorphic chips. There's a disconnect:
SNNs are typically trained on GPUs anyway (as authors acknowledge)
The pruning ratios used during training may not reflect deployment scenarios
How does training with high pruning ratios affect actual neuromorphic deployment efficiency?

Performance degradation concerns:
On some tasks, there are non-trivial accuracy drops (e.g., 1.3% on SpikformerV2-8-384 T=4, 2% on QKFormer variants)
On COCO object detection, the 1.5% mAP drop is dismissed as "acceptable," but this is subjective
The paper doesn't discuss scenarios where DSTSP might fail or provide guidelines for when not to use it

Writing and presentation:

The term "Spiking-token Mixer" vs "STMixer" vs "SMixer" is used inconsistently
Figure 5 could be clearer—the temporal pruning illustration is hard to parse
Some claims lack support: "The intrinsically low firing rates of SNNs result in significant network redundancy" (line 83)—citation needed

**Questions:**

See weakness

---

> ### Author Response · Authors · 2025-11-20
> **Rebuttal**
>
> Thanks for your valuable comments. We will explain and discuss your concerns.
>
> ### Weaknesses:
> > ***Weakness 1**: Limited novelty of SMixer architecture: The Spiking-token Mixer concept appears to be directly adapted from STMixer (Deng et al. 2024). The main architectural contribution seems incremental—demonstrating that this existing architecture can replace SSA in other frameworks. The novelty primarily lies in the pruning framework rather than the architecture itself.*
>
> **A1**: Theoretically, we regard pruning and model architecture as an integrated unity. This study establishes a high-performance, event-driven SNN paradigm and investigates training acceleration strategies for practical deployment, aiming to balance high accuracy with computational efficiency across various tasks. In our experiments on the ImageNet dataset, we first validated the scalability of the STMixer. By replacing the Spiking Self Attention (SSA) mechanism in state-of-the-art Spiking Transformer architectures with the Spiking Token Mixer paradigm, we observed no degradation in performance, thereby substantiating the scalability of STMixer. Furthermore, the subsequent application of DSTSP significantly accelerated training and reduced energy consumption while maintaining inference accuracy with only small degradation.
>
> > ***Weakness 2**: The DSTSP framework is designed to reduce training overhead on GPUs, but the motivation emphasizes event-driven deployment on neuromorphic chips. There's a disconnect: SNNs are typically trained on GPUs anyway (as authors acknowledge) The pruning ratios used during training may not reflect deployment scenarios How does training with high pruning ratios affect actual neuromorphic deployment efficiency?*
>
> **A2**: We argue that aggressive pruning is necessary to mitigate the training latency bottleneck inherent to SNNs on GPUs compared to ANNs. Moreover, the STMixer architecture is specifically optimized for neuromorphic deployment, providing a distinct efficiency advantage over SSA. Thus, the main contribution of this paper is to validate the full-stack feasibility of the STMixer paradigm, rigorously exploring its viability from the training phase through to inference
>
> > ***Weakness 3**: Performance degradation concerns: On some tasks, there are non-trivial accuracy drops (e.g., 1.3% on SpikformerV2-8-384 T=4, 2% on QKFormer variants) On COCO object detection, the 1.5% mAP drop is dismissed as "acceptable," but this is subjective The paper doesn't discuss scenarios where DSTSP might fail or provide guidelines for when not to use it.*
>
> **A3**: We acknowledge the performance compromise, particularly in fine-grained tasks such as object detection. A potential limitation of DSTSP arises in scenarios like COCO object detection, where achieving maximal precision necessitates hierarchical downsampling at original resolutions, resulting in variable token counts. Our method mitigates this by enforcing token uniformity via interpolation, which inherently reduces precision. Consequently, the deployment of DSTSP is not recommended when computational resources are abundant. In such cases, specifically for fine-grained tasks with limited fine-tuning epochs, employing the full-scale STMixer is preferable. Future work will focus on exploring high-precision pruning methodologies specifically tailored for fine-grained vision tasks.
>
> > ***Weakness 4**:The term "Spiking-token Mixer" vs "STMixer" vs "SMixer" is used inconsistently Figure 5 could be clearer—the temporal pruning illustration is hard to parse Some claims lack support: "The intrinsically low firing rates of SNNs result in significant network redundancy" (line 83) --citation needed*
>
> **A4**: Sorry for the confusion. Spiking-token Mixer and STMixer refer to the mechanism and the full architecture from [1], respectively, while SMixer denotes the baseline for our DSTSP method. In the temporal figure, the bar visualization signifies the SIV statistic for a given time step, used to select high-information steps for retention. This illustration will be refined in the final manuscript.
>
> [1] Spiking Token Mixer: An Event-Driven Friendly Former Structure for Spiking Neural Networks. NIPS 2024

---

> ### Comment · Reviewer_B5FE · 2025-11-28
>
> I would like to express my gratitude to the authors for their comprehensive response and the significant effort put into the additional experiments. The revisions have successfully clarified several key aspects of the proposed approach. However, I still have two remaining concerns:
>
> 1. Could you provide comparisons with other state-of-the-art models or methods to further verify the robustness of Smixer?
>
> 2. I noticed that DSTSP outperforms Smixer in certain scenarios within the time series tasks. Could you elaborate on the potential reasons for this phenomenon?

---

> > ### Author Response · Authors · 2025-11-30
> > **Further Response**
> >
> > We now address each of your questions.
> >
> > **C1**: First, we present a comparison between our method and SNN weight pruning techniques in Table 7 in the manuscript, demonstrating the effectiveness of our approach. Furthermore, we include a comparison with the latest pruning methods associated with Spikformer, applied to the Spiking Token-mixer architecture. As shown in the table below, the results on the CIFAR100 highlight the robustness of our method.
> >
> > #### Table R1-3
> > | Pruning Method  |Basic Acc(%) | Pruned Acc(%) |
> > | -------- |:--------:|:--------:
> > | DSTSP（ours）   | 81.87  |  81.03 |
> > | STATA [1]   |81.87  |  80.12  |
> >
> >
> >
> > **C2**: We posit that the extensive spatiotemporal redundancy inherent in temporal tasks frequently dilutes valuable information. By effectively pruning this redundancy to retain only key temporal information, our DSTSP notably enhances performance across several datasets.
> >
> > [1] Towards Efficient Spiking Transformer: a Token Sparsification Framework for Training and Inference Acceleration. ICML2024

---

### Official Review · Reviewer_sSmP · 2025-10-31

**Soundness:** 3
**Presentation:** 3
**Contribution:** 3
**Rating:** 6
**Confidence:** 5

**Summary:**

The paper proposes SMixer, an event-driven SNN architecture that replaces spiking self-attention with a Spiking-token Mixer to improve hardware feasibility while retaining competitive accuracy. Building on SMixer, the authors introduce Dynamic Spatial-Temporal Spiking Pruning (DSTSP)—a parameter-free framework that exploits sparsity in spike features by (i) pruning low-information tokens in space based on a Spike Intensity Value (SIV) metric and (ii) reducing uninformative time steps and spike counts in time, thereby cutting training overhead without materially degrading performance. Across ImageNet-1K, CIFAR-10/100, DVS datasets, time-series forecasting, and COCO detection, the approach demonstrates substantial reductions in memory/energy and higher throughput, with accuracy maintained close to baselines; in some ImageNet settings it even slightly improves top-1. The work argues for SNN designs that are simultaneously fully event-driven, low-overhead to train, and competitive in performance, positioning SMixer + DSTSP as a practical blueprint for neuromorphic deployment.

**Strengths:**

1. Clear problem framing and design criteria. The paper precisely motivates why spiking self-attention is ill-suited to asynchronous chips and formalizes three desiderata—fully event-driven operation, low training overhead, and competitive performance—guiding the architectural choices.

2. Hardware-friendly mixer in place of SSA. Replacing spike-matrix multiplications with a learnable mixer weight (WM) preserves the benefits of token mixing while avoiding timing-sensitive QK interactions, improving deployability on neuromorphic hardware.

3. Parameter-free pruning mechanism. DSTSP is simple (addition/sorting), incurs negligible extra compute, and integrates into training without auxiliary modules or epochs; pruned features also skip back-prop, directly lowering cost.

4. Principled redundancy metric (SIV) with empirical support. The SIV-based selection aligns with semantic saliency: high-SIV regions correspond to foregrounds and yield higher accuracy than random or high-SIV pruning baselines.

5. Comprehensive experiments and tangible efficiency gains. On ImageNet, SMixer and SMixer + DSTSP achieve comparable accuracy to Spiking Transformers while reducing memory and energy and boosting throughput; similar trends hold on CIFAR/DVS, time-series, and COCO detection.

6. Analytical backing. The paper provides operation and energy models showing why spatial-temporal pruning reduces cost relative to the original mixer; it also contrasts DSTSP with weight-pruning methods on cost/accuracy/sparsity.

**Weaknesses:**

1.The robustness of the SIV (Social Influence Value) model under adversarial inputs or input perturbations has not been thoroughly investigated. This limitation raises concerns regarding the reliability of SIV in real-world applications, where data quality and integrity cannot always be guaranteed.
2.While pruning effectively reduces model complexity and inference costs, it inherently leads to a reduction in model capacity. The extent to which this capacity loss can be mitigated by simply increasing the number of training epochs remains underexplored. Without a clear understanding of this trade-off, it is uncertain whether prolonged training truly restores the representational power of the pruned model or merely overfits to the remaining parameters.
3.The paper does not clarify why DSTSP is intrinsically incompatible with the SSA architecture.
4.DSTSP’s efficacy is only validated on small neuromorphic datasets; its scalability to large-scale, real-world benchmarks like HardVS remains untested.

**Questions:**

1.Give more ablation on robustness of SIV.
2.Does increasing the number of training epochs adequately compensate for the capacity loss induced by pruning?
3.Please further explain why DSTSP is uniquely suited for the SMixer architecture rather than the SSA architecture.
4.Provide experimental results on HarDVS dataset.

---

> ### Author Response · Authors · 2025-11-20
> **Rebuttal**
>
> Thank you for your detailed comments and suggestions for improvement. We would like to address your concerns and answer your questions below.
>
> ### Weaknesses and Questions:
> > ***Weakness 1, Question 1**: The robustness of the SIV model under adversarial inputs or input perturbations has not been thoroughly investigated. Give more ablation on robustness of SIV.*
>
> **A1**: Following your suggestion, we further investigate the robustness of the SIV module by introducing Gaussian noise perturbations ( $\sigma=0.2$ ) to the input data. As presented in Table R1-1, we conduct comparative experiments on both the static dataset CIFAR100 and the neuromorphic dataset CIFAR10-DVS, evaluating accuracy across different configurations (i.e., with and without noise during both training and inference phases). The results demonstrate that our SIV-based model achieves accuracy comparable to the noise-free baseline in all scenarios, thereby validating the robustness of the SIV mechanism.
>
> #### Table R1-1
> | Add Noise  |CIFAR100 | CIFAR10-DVS |
> | -------- |:--------:|:--------:|
> | None    | 81.03  |  82.6  |
> | Train   | 80.97  |  82.2  |
> | Inference| 80.57 | 82.0  |
> | Train+Inference| 80.81 | 82.3   |
>
>
> > ***Weakness 2, Question 2**: It remains unclear whether extending training can fully recover this lost representational power or if it merely leads to overfitting, a trade-off that is currently underexplored.*
>
>
> **A2**: Thank you for your question. First, we maintain the same number of training epochs as the original model, which demonstrates the inherent utility of our pruning strategy. Second, we provide results across varying epoch counts in Table R1-2. These results indicate that altering the number of epochs does not effectively mitigate the performance degradation caused by pruning. This suggests a negligible correlation between the epoch hyperparameter and the model's performance, further validating the effectiveness of our proposed pruning method.
>
> #### Table R1-2
> | Architecture  |Epoch | Acc（top-1） |
> | -------- |:--------:|:--------:|
> | STMixer    | 300  |  76.68  |
> | STMixer + DSTSP    | 200  |  76.69  |
> | STMixer + DSTSP    | 300  |  76.87  |
> | STMixer + DSTSP    | 400  |  76.89  |
> | SpikformerV2    | 300  |  80.38  |
> | SpikformerV2 → M+ DSTSP   | 200  |  79.03 |
> | SpikformerV2 → M+ DSTSP   | 300  |  79.15  |
> | SpikformerV2 → M+ DSTSP   | 400  |  79.21  |
>
>
>
> > ***Weakness 3, Question 3**: Please further explain why DSTSP is uniquely suited for the SMixer architecture rather than the SSA architecture.*
>
> **A3**: Thank you for your question. In the spatial pruning phase, we prune a subset of weights within the linear layers of the Token Mixer by directly sparsifying the $N \times N$ learnable attention map, thereby reducing both parameter complexity and computational overhead. In contrast, the $N \times N$ attention weights in Spiking Self-Attention (SSA) are derived dynamically from the matrix multiplication of the query and key feature matrices. Consequently, the underlying mechanisms of these two approaches are fundamentally distinct. Therefore, the proposed DSTSP is specifically tailored for the STMixer architecture.
>
>
> > ***Weakness 4, Question 4**: Provide experimental results on HarDVS dataset.*
>
> **A4**: Thank you for your advice.  We provide results on HarDVS dataset in Tab. R1-3.
>
> #### Table R1-3
> | Architecture  |Param(M) | Power(mJ) | Step | Acc(%) |
> | -------- |:--------:|:--------:|:--------:|:--------:|
> | SDT-V2   | 18.3  |  8.0  |  8 $\times$  1| 47.5 |
> | SDT-V3   | 18.7  |  18.1  |  8 $\times$  4| 48.9 |
> | SMixer   | 17.9  |  17.8  |  8 $\times$  4| 49.1 |
> | SMixer + DSTSP   | 16.7  |  8.9  |  8 $\times$  4| 48.2 |

---

> > ### Comment · Reviewer_sSmP · 2025-11-24
> >
> > Thank you for your response. My concerns have been addressed and I will adjust my score accordingly.

---

### Author Response · Authors · 2025-12-03
**Summary**

For the convenience of the AC, we have summarized the reviewers' feedback and our responses as follows.
### Strengths

**Clear Problem Motivation & Framing:** All reviewers acknowledge that the paper addresses critical issues in current SNN research, specifically the incompatibility of Spiking Self-Attention (SSA) with asynchronous hardware and the high computational cost of training. The motivation to replace SSA with the hardware-friendly Spiking token mixer paradigm and the corresponding DSTSP pruning framework is recognized as well-grounded, effectively addressing the critical bottlenecks of high training overhead and deployment latency.

**Efficiency & Practicality:** The proposed DSTSP method is widely praised for being parameter-free, lightweight. Reviewers noted significant reductions in memory usage, training time and energy consumption (Reviewers sSmP, B5FE, 2HdN) and its ability to integrate without auxiliary training modules (Reviewer sSmP).



### Questions

**Methodology & Mechanism Clarifications:**

Reviewer sSmP requested clarification on the intrinsic incompatibility between DSTSP and the SSA architecture, and the rationale behind its unique suitability for SMixer. We noted that in the spatial pruning phase, we pruned a subset of weights within the linear layers of the Token Mixer by directly sparsifying the $N \times N$ learnable attention map, thereby reducing both parameter complexity and computational overhead.

Reviewers B5FE and tRYC invited us to further elaborate on the distinctive architectural features of our model relative to prior work, and to clarify how the DSTSP framework operates within the asynchronous inference dynamics of SNNs. We treated pruning framework and STMixer architecture as an integrated whole, establishing a high-performance, event-driven SNN paradigm that balances accuracy and computational efficiency. We first validated the scalability of the STMixer by replacing the Spiking Self-Attention (SSA) in state-of-the-art Spiking Transformers, achieving comparable performance without degradation. Building on this, the DSTSP significantly accelerates training and reduces energy consumption, with only minimal accuracy loss during inference. Crucially, pruning in DSTSP is a training-time strategy. During inference on neuromorphic hardware, efficiency stems from the inherent sparsity and compatibility of the STMixer framework, not from explicit pruning.


**Clarity of Figures and Metrics:**

Reviewers 2HdN & tRYC both requested detailed explanations for Figure 4 (specifically the vertical axis, RIE, and MS-SSIM metrics) and found Figure 6 (visualization of pruning/saliency) unpersuasive or unclear regarding background noise. We have provided the corresponding clarifications and updated the visualization results in Figure 6.

**Performance vs. Efficiency Trade-off:**

Reviewer B5FE and tRYC noted non-trivial accuracy drops (e.g., on COCO) that were subjectively dismissed as "acceptable." We characterize the proposed SMixer and DSTSP as an initial investigation into efficient SNN architectures that span from training acceleration to hardware deployment. Although the method achieves competitive performance with marginal loss across various domains, we acknowledge limitations in high-precision tasks like object detection. The current framework’s fixed-resolution constraints require operations such as pooling, leading to information loss. We intend to rigorously investigate and mitigate this issue in future research.

**Disscussions on Hardware Deployment:**

Reviewer 2HdN & tRYC requested discussions on edge-hardware deployment. We noted that we advanced the initial findings of Spiking token mixer by realizing a complete training-to-inference pipeline for STMixer. The STMixer has been previously demonstrated the effectiveness through experimental deployment on neuromorphic hardware. This constitutes an initial yet concerted exploration of on-chip deployment, explicitly tailored to the stringent constraints of asynchronous neuomorphic hardware.

---

### Meta-Review · Area_Chair_zUbV · 2025-12-21

**Summary:**

This paper presents SMixer, an event-driven spiking neural network (SNN) with a hardware-friendly Spiking Token Mixer replacing the spiking self-attention (SSA) and a Dynamic Spatial-Temporal Spiking Pruning (DSTSP) framework, greatly alleviating the time lag in training. It also tackles two primary issues in currentcd SNNs - (1) The non-match of SSA with asynchronous neuromorphic hardware, and (2) Computation and memory expensive GPU training. The authors present a computing flow that the SNNs are complete event-driven, with training efficiency and accuracy on par with some power hungry models, where SMixer+DSTSP is an instantiationn.

Reviewers rate this paper as a well-motivated, realistic and comprehensive work. Reviewers believe this paper focuses on efficiency, deployability and the lower cost of training. Orchestrator novelty, mechanism- and metricc collinearity, performance-efficiency trade-offs, pruning criterion generalization robustness and lack of direct neuromorphic hhardware validation were the primary concerns. The authors wrote a strong rebuttal and introduced new experiments (HarDVS, robustness), explanations, improved their error analysis section and described short-comings and use-cases. Several reviewers raised scores post-rebuttal.

**Reviewer Concerns:**

Shared comments:

1. Performance and Efficiency Trade-off. Most reviewers said DSTSP provides high training efficiency, memory, and energy gains but causes non-trivial accuracy drops on some tasks (e.g., COCO detection, some Transformers). They also asked about justification. Resolution: Authors clarified the method targets on efficiency-critical scenarios, admitted not ideal for high-precision tasks with ample resources. They also discussed failure modes (e.g., fixed token resolution, interpolation) and claimed it as an initial steps.

2. Clarity of Methodology. Reviewers requested clearer definitions of metrics (e.g., RIE, MS-SSIM), pruning visuals, axis labels, table annotations, and consistent terminology (SMixer vs. STMixer vs. Spiking-token Mixer). Resolution: Authors added formal definitions in appendix, clarified figure intuitions and axes, corrected mislabled tables, fixed accuracy bug, and committed to better visuals and terminology.

3. Justification of DSTSP. Reviewers asked why DSTSP suits SMixer over SSA-based architectures, and if it preserves event-driven inference. Resolution: Authors explained DSTSP exploits static, learnable mixing weights for direct sparsification, unlike SSA's dynamic QK interactions incompatible with pruning. They said DSTSP is training-time only, avoiding inference-time operations violating asynchronous executionn.

4. Hardware Deployment. Concerns over lacking direct neuromorphic hardware experiments. Resolution: Authors said this builds on prior STMixer hardware deployments, providingg full training-to-inference pipeline analysis.

Reviewer-Specific Questions:

1. Reviewer sSmP: SIV criterion robustness, pruning capacity loss, SSA compatibility, and HarDVS absence. Authors: noise robustness tests, epoch-scaling studies, HarDVS evaluations, and DSTSP specificity explanation. Reviewer resolved concerns and raised scores.

2. Reviewer B5FE: novelty, training-inference motivation mismatch, performance drops, writing clarity. Authors: full-stack paradigm, DSTSP usage, pruning comparisons, limitation acknowledgment. Concerns narrowed, partially resolved via discussions.

3. Reviewer 2HdN: hardware experimenbts lack and unclear figures. Authors: prior hardware links, improving explanations. The reviewer resolved doubts and increased score.

4. Reviewer tRYC: metric definitions, efficiency claims, incorrect reporting, inference behavior, visualization noise. Authors correcteds errors, added tables, stated DSTSP non-inference. Reviewer raised score, supported acceptance.

**Reviewer Scores:**

The scores of all reviewers are reasonable. Multiple reviewers have raised their score after the rebuttal.

---

### Decision · Program_Chairs · 2026-01-26

Accept (Poster)